# Role of Endoscopy in Esophageal Tuberculosis: A Narrative Review

**DOI:** 10.3390/jcm11237009

**Published:** 2022-11-27

**Authors:** Tong Ye, Ye Zong, Guiping Zhao, Anni Zhou, Bing Yue, Haiying Zhao, Peng Li

**Affiliations:** Department of Gastroenterology, Beijing Friendship Hospital, Capital Medical University, Beijing 100050, China

**Keywords:** endoscopy ultrasonography, contrast-enhanced harmonic endoscopic ultrasonography, elastography, endoscopic ultrasound-guided fine-needle aspiration, esophageal tuberculosis

## Abstract

Esophageal tuberculosis (ET) is a rare infectious disease of the gastrointestinal tract. Awareness of ET is deficient due to its low incidence. Unexplained dysphagia and upper gastrointestinal bleeding are the most common symptoms of ET. The prognosis is generally good if patients are diagnosed properly and receive anti-tubercular treatment promptly. However, ET is difficult to differentiate from other diseases. Endoscopic techniques such as esophagogastroduodenoscopy (EGD), endoscopic ultrasonography (EUS), contrast-enhanced harmonic endoscopic ultrasonography (CH-EUS), elastography, and endoscopic ultrasound--guided fine-needle aspiration (EUS-FNA) improve the diagnosis of ET. Thus, the characteristics of ET and other difficult-to-detect diseases according to EGD and EUS were summarized. Intriguingly, there is no literature relevant to the application of CH-EUS and elastography in ET. The authors’ research center was first in introducing CH-EUS and elastography into the field of ET. The specific manifestation of ET based on CH-EUS was discovered for the first time. Correlative experience and representative cases were shared. The role of endoscopy in acquiring esophageal specimens and treatment for ET was also established. In this review, we aim to introduce a promising technology for the diagnosis and treatment of ET.

## 1. Introduction

Esophageal tuberculosis (ET) is a rare disease, accounting for around 0.15% in the necrotic tissue of subjects with tuberculosis and about 0.07–3.00% for subjects with gastrointestinal tuberculosis [1,2]. To date, with the resurgence of tuberculosis, the incidence of ET has gradually increased [3]. However, the diagnosis of ET remains challenging because its clinical features are often nonspecific and variable. It may be mistaken for esophageal carcinoma and could lead to unnecessary surgery [2]. Following a definite diagnosis and treatment with anti-tuberculosis therapy, the patient will become asymptomatic after 6 months and obtain endoscopy healing at 1-year follow-up [2,3,4,5,6], which is widely divergent from malignancy. Despite the optimistic prognosis of ET, it could become life-threatening in some cases, for example, massive bleeding resulting from aortoesophageal fistula [5]. Therefore, prompt and explicit diagnosis has always been the essential and challenging aspect of ET.

Esophagogastroduodenoscopy (EGD) could observe the morphology of the lesion directly. Endoscopic ultrasonography (EUS) can detect the echogenic characteristics of the lesion, different layers of the esophageal wall, and para-esophageal organs and tissues. Additionally, EUS is able to guide fine-needle aspiration to obtain extra-esophageal tissue such as an infected lymph node [7]. Contrast-enhanced harmonic endoscopic ultrasonography (CH-EUS) delineates the vascular distribution and blood perfusion of the target organs via an agent injected into the superficial median cubital vein. The hemodynamic state of blood vessels is described by CH-EUS with high resolution, which is helpful in distinguishing benign from malignant [8,9]. Elastography has been developed as a qualitative and quantitative technique for the assessment of elasticity of different tissues. Previous studies revealed the advantage of elastography in the differential diagnosis of lymph nodes [10,11,12].

Thus, in this article, we summarize recent literature on the application of endoscopy techniques in ET, share several representative cases, and highlight the experience of diagnosis of ET in our research center.

## 2. Classification and Clinical Manifestation

According to absence or presence of extraesophageal tubercular lesions, tubercle bacilli’s involvement in the esophagus can be classified as primary or secondary [13]. The differences between primary and secondary esophageal tuberculosis are listed in Table 1. The majority of ET is secondary in adjacent tuberculosis lesions, such as extension of mediastinal lymph nodes, pulmonary, laryngeal, or Pott’s spine or, less commonly, through hematogenous spread [3,14,15,16,17]. Primary ET occurs when patients swallow sputum or food contaminated with tubercle bacilli. However, primary ET very rarely occurs due to multiple efficacious esophageal protective mechanisms such as stratified squamous epithelium, saliva, and mucus [18]. 

The symptoms of ET are variable and depend on the endoscopic morphology. Dysphagia is the main symptom and can be caused by several factors, such as obstruction of intrinsic pseudotumor on account of fibrosis formation or extrinsic compression of infected mediastinal lymph nodes. Upper gastrointestinal tract hematemesis is often caused by damage of blood vessels at the base of an ulcer or artery–esophageal fistula. Respiratory symptoms such as cough and wheezing are the result of a tracheoesophageal fistula. Anorexia, fatigue, night sweats, low-grade fever, and weight loss are common tuberculemia symptoms [19,20,21,22,23,24,25,26,27].

## 3. Manifestations of ET According to Upper GI Endoscopy

ET has multiple morphological types according to EGD which can be summarized as follows: ulcerations, eminence lesions, fistula, stricture, and traction diverticula (Table 2). The most common morphology is mid-esophageal linear ulcerations with irregular infiltrated margins and grayish membranous necrotic base [17,22,28]. ET ulcerations usually occur in the mid-esophagus. Tuberculosis-infected lymph nodes are primarily located in the subcarinal region and always intrude into the esophagus on the same level and therefore result in the formation of mid-esophageal ulcerations [21,28]. Deep and large ulcerations have a bleeding tendency and usually present with recent petechiae. Sometimes, ulcerations invade the aorta and start a lethal hemorrhage [5,17]. When encountering unexplained upper gastrointestinal bleeding, the possibility of ET should also be taken into consideration. Nevertheless, the specificity of morphology was inferior; when ulcerations are observed in the upper or middle esophagus, in addition to ET, Crohn’s disease and Behçet’s disease should also be considered. To differentiate these diseases, other symptoms should be taken into account. For instance, Crohn’s disease has representative ileocecal longitudinal, discontinuous, and cobblestone ulcerations [29] and extra-intestinal damage in joints, skin, and eyes, or oral mucosa [30]. Behçet’s disease causes ulcers in the mouth, eyes, and genitals [31,32].

Eminence lesions comprise intrinsic protruding lesions and extrinsic bulge compression (Figure 1a and Figure 2a). The surface of eminence lesions can be smooth, with ulcerations, fistula, or diverticula [6,18]. Using EGD, the granular form appears as scattered verrucous grayish nodules accompanied by ulcers or erosions. The hypertrophic form refers to when the esophageal wall is fibrotic and forms a pseudotumor [33]. Carina lymph nodes are deposited under trachea bifurcation, close to the mid-esophagus. In addition to causing ulcers, carina lymph nodes also compress the esophagus, and present as extrinsic eminence [21]. Moreover, mediastinal fibrosis can compress the esophagus and lead to extrinsic bulges or traction diverticula [34]. Once protruding lesions block and cause the stricture of the lumen, patients will experience varying degrees of dysphagia, especially with confusing symptoms such as weight loss, making differential diagnosis from malignancy more difficult.

Tuberculous lesions encroach on and result in the formation of an abnormal communication between the esophagus and trachea. A double-barreled appearance can be seen when using endoscopy (Figure 3a,b). Meanwhile, the fistulous opening can be observed near the carina using bronchoscopy. A tracheoesophageal fistula can increase the risk of aspiration pneumonia, as reported previously [5]. This double-barreled appearance may resemble fistula tracts caused by aggressive fungal infection or esophageal carcinoma. Histopathology and pathogeny can be relied on to confirm the diagnosis [35,36].

## 4. Manifestations of ET According to EUS

EUS has unique advantages in the diagnosis of ET [17,18] (Table 2). It can depict the echogenic characteristics of a tuberculosis lesion, different layers of the esophageal wall, and periesophageal tissue, and the relationship between them. Moreover, endoscopic ultrasound-guided fine-needle aspiration (EUS-FNA) can acquire deep biopsy tissues for further examination [37].

In most cases, ET is caused by the rupture of infected mediastinal lymph nodes to the esophagus. The esophageal wall thickens, and the boundary between the surrounding lesions becomes vague [17,18].

The features of a tuberculosis lymph node when using EUS depend on its stage of development [17]. Lymphoid hyperplasia is the first phase when lymph nodes try to clear the invading mycobacterium tuberculosis via lymphoid tissue hyperplasia. In this phase, hyperplastic lymphocytes and a small amount of caseation concurrence are shown as homogeneous and hypoechoic by EUS. Then, caseous necrosis is further aggravated and leaves hyperechoic foci or strands on heterogeneous hypoechoic background which represent calcification or fibrosis, respectively. Moreover, the adventitia of the lymph nodes is destroyed, and lymph nodes fuse with each other, which appears as indistinct adventitia with partial matting. Overall, based on EUS, the characteristics of lymph nodes can be summarized as heterogeneous, predominantly hypoechoic mass with local hyperechoic foci or strands, indistinct adventitia, and occasionally partial matting (Figure 1b, Figure 2b and Figure 3c) [1,6,18].

Both ET and carcinoma manifest as hypoechoic masses encroaching into various layers of the esophageal wall. Nevertheless, esophageal carcinoma is a hypoechoic mass derived from the epithelial layer, infiltrating from the inside to the outside [38,39]. Most ET is secondary to mediastinal lymph nodes, which are invaded externally. Benign esophageal submucosal tumors (SMTs) are easily identified from ET. Benign esophageal SMTs, such as esophageal leiomyoma, originate from the esophageal lamina propria or mucosal muscle layer with homogeneous hypoechoic lesion, regular boundaries, and normal mediastinal lymph nodes. Furthermore, the echo of malignant esophageal SMTs, such as leiomyosarcoma, is heterogeneous hypoechoic with irregular margins and malignant lymph nodes [40,41,42]. Malignant lymph nodes present as spherical, hypoechoic, and sharply demarcated [42].

Sarcoidosis is a non-caseous necrotic granulomatous lesion involving multiple organs and systems, particularly affecting both lungs, and hilar and mediastinal lymph nodes. Some experts have proposed that sarcoidosis could be easily differentiated from ET by using EUS because sarcoidosis-associated mediastinal lymph nodes are isoechoic, well margined, and clustered [43,44]. Other experts have suggested that EUS findings for sarcoidosis-associated lymph nodes were nonspecific [45].

The incidence of esophageal Crohn’s disease is 0.3–10% [46]. It is difficult to distinguish Crohn’s disease from ET using EUS [46]. Only one case regarding Crohn’s disease reported that the esophageal wall thickened to 15.4 mm and presented as heterogeneous echo with strand-like hyperechoic areas using EUS. Five layers of the esophageal wall were destroyed. The extra-esophageal lymph nodes were hypoecho and enlarged to 3–4 mm [47]. Differential diagnosis should also refer to whether patients have typical ulcers in the ileocecum or whether there are extra-intestinal manifestations. Unfortunately, there is nothing in the literature on the application of EUS in Behçet’s disease.

## 5. Manifestations of ET According to CH-EUS and Elastography

CH-EUS was performed in two cases in our research center. Both lesions featured hypo-enhancement compared with the surrounding tissues (Figure 1b and Figure 2b) (Table 2). Mechanically, hypo-enhancement reflects a deficiency of blood supply, which corresponds to Stage II or above when the lymph nodes undergo caseous necrosis. Caseous granuloma is always detected in infected lymph nodes which are in Stage II or a more severe stage. In fact, granuloma was found in Cases 1–4 which we will introduce later. Using elastography, images of two lesions were mainly green, showing that both lesions were soft (Figure 1c and Figure 2c) (Table 2). The characteristics of ET when using CH-EUS and elastography are worth further evaluation.

## 6. Role of Endoscopy in Acquiring Esophageal Specimens

A definite diagnosis of ET ultimately relies on pathologic and pathogenic investigations of the esophageal specimens. Common methods to obtain biopsy specimens include using biopsy forceps to remove regional esophageal mucosa and using EUS-FNA to obtain a strip of tissue from deep tissues (Figure 1 and Figure 2d) (Table 3). The specimens acquired by biopsy forceps or EUS-FNA are conserved in formalin. The first choice is acquiring esophageal mucosal tissues with biopsy forceps. The positivity rate may be elevated by multi-point deep biopsy. Obtaining samples at the edge of the ulceration is secure and sufficient. Indications of EUS-FNA are listed as follows: 1. The lesion originates from the submucosa or outside the digestive tract; 2. Routine biopsy by forceps was negative, but ET is still highly suspected.

The histopathological specialty of typical tuberculosis granuloma shows central caseous necrosis, surrounded by a radial arrangement of epithelioid cells, with scattered Langhans giant cells. Lots of lymphocytes and fibrous connective tissue can be viewed around the nodule, whereas classical tuberculosis granuloma is uncommon in clinical practice, the emergence of epithelioid cells, multinucleated giant cells along with caseous necrosis is sufficient for diagnosis [25,48]. One study proposed that even if there was no caseous necrosis, ET could be diagnosed [28]. In addition, other than tuberculosis granuloma, Crohn’s disease, sarcoidosis, syphilis, and fungal infections also manifest as granuloma. Consequently, it is still difficult to differentiate ET from other granulomatous diseases [26,49,50]. Endoscopy techniques could provide more information for definite diagnosis of ET.

Taking secretions from the center of the ulcer may be beneficial to the detection of pathogens. Ziehl–Neelsen staining is recommended to discover acid-fast bacilli (AFB). To our knowledge, Ziehl–Neelsen staining has a limited sensitivity between 40–60% [51,52,53], and only shows positive in <25% patients [54,55]; furthermore, 75% cases with extrapulmonary disease could not recognize acid-fast bacilli [53]. Cultures for AFB enjoyed an impressive sensitivity and specificity, recorded as 80% and 98%, respectively. However, it takes a great deal of time, about 6 to 8 weeks, to cultivate mycobacterium tuberculosis [51,52,53]. The molecular biology technique such as polymerase chain reaction (PCR) is reliable and worthwhile in the diagnosis of ET [3,5,48]. PCR is faster than culture, and the sensitivity of PCR is similar to culture [48,56]. Forceps biopsy and EUS-FNA provide sufficient histological specimens for PCR.

**Table 3 jcm-11-07009-t003:** Role of endoscopy in acquiring esophageal specimens and different testing methods.

Classification	Function	Testing Methods		Reference
Biopsy forceps	Removing regional esophageal mucosa	Histopathology: tuberculosis granuloma		[25,28,48]
		Etiology	Ziehl-Neelsen staining	[51,52,53,54,55]
EUS-FNA	Obtaining strip of tissue from deep tissues		Cultures	[51,52,53]
			Polymerase chain reaction	[3,5,48,56]

EUS-FNA: endoscopic ultrasound-guided fine-needle aspiration.

## 7. Medication and Endoscopic Treatment

A standard nine-month course of treatment consists of four drugs (isoniazid, rifampicin, ethambutol, and pyrazinamide) for two months and two drugs (isoniazid and rifampicin) for seven months. Multidrug resistance of mycobacterium tuberculosis has gradually been acquired increasing attention from the scientific community. Biopsy forceps and EUS-FNA are expected to acquire biopsy specimens for multidrug-resistance detection. For example, rolling circle amplification (RCA) is a fast, highly sensitive, and highly specific molecular biology technique (96.6% and 89.5%, respectively) for detecting multidrug-resistant mycobacterium tuberculosis. The identification of multidrug-resistant bacteria will improve therapeutic efficacy and optimize treatment outcomes [57,58,59].

If traditional drug treatment is not effective, endoscopic intervention such as the closure of fistula can be performed. Recently, an associated case illustrated successful closure of the fistula with a 12 mm over-the-scope clip (OTSC) [60]. A cap was attached to the tip of the endoscope; then an OTSC was deposited on the tissue in the cap, and a vacuum was generated through endoscopic suction to pull the tissue around the fistula into the cap. The success rate of OTSC was 33–77%. [61,62,63]. The epithelialization of the fistula is speculated to make it difficult to heal. Argon plasma coagulation (APC) can be utilized to burn the epithelialized mucosa before clipping, which may be more conducive to the healing of the fistula [64] (Table 4).

## 8. Case Reports

We retrospectively analyzed medical records from 2013 to 2022 from Beijing Friendship Hospital and screened a total of four patients who were diagnosed with ET (Table 5). The chief complaint in all patients was dysphagia. None of them had AIDS, syphilis, hepatitis, post-transplantation status, or other diseases that led to being immunocompromised. CH-EUS and elastography was performed in two cases. Enlarged mediastinal lymph node and thickening of the middle esophageal wall were detected in all of the cases, which were considered as secondary ET. There were two cases that presented as eminence lesions. (1) A 1.5 × 1.5 cm protruding lesion was found in Case 1, which was 28 cm from the incisor. The surface mucosa of the lesion was uneven with central depression (Figure 1a). EUS showed that the mediastinum mass was mainly hypo-echoic mixed with hyper-echo, and the boundary between the mass and the esophageal wall was vague. Using CH-EUS, it was found that the mass was hypo-enhanced (Figure 1b). The lesion was soft according to elastography (Figure 1c). Then, EUS-FNA was performed (Figure 1d). (2) An elevated submucosal lesion with fissure-like changes in the surface was found in Case 2 (Figure 2a). The lesion was 0.8 × 0.8 cm and 25 cm from the incisor. EUS indicated that the hypoechoic lesions violated the esophageal wall (lymph nodes were considered). CH-EUS also showed hypo-enhancement (Figure 2b), and elastography revealed that the lesion was soft (Figure 2c). EUS-FNA was performed to puncture the extra-esophageal hypoechoic lesions from the mid-esophagus (Figure 2d). (3) Multiple ulcers with esophageal stricture were shown in Case 3. Scattered ulcers were detected in the esophageal mucosa within 18 cm of the incisor with white moss on the bottom. Then, we carefully passed through the stricture using an extra slim scope and circumferential mucosal ulcers with nodular changes extending to the cardia emerged. (4) A 1.0 × 2.5 cm longitudinal fistula with pus overflowing in the anterior esophageal wall at 27–29 cm from the incisor was demonstrated in Case 4. A granule-like tissue was seen at the base of the fistula, and no lichen was attached at the base. The surface of the fistula and the surrounding mucosa were red (Figure 3a,b). EUS revealed a heterogeneous hypoechoic background with hyperechoic strands which were detected under the carina and were fused with each other (Figure 3c).

The pathological results discovered granuloma in all patients. PCR and Ziehl-Neelsen staining were performed on two patients, but the results were all negative. Finally, diagnostic anti-tuberculosis therapy was conducted. After 9 months of drug treatment, four patients were asymptomatic. After 1-year follow-up, all of the lesions had disappeared and left scars on the esophageal mucosa according to endoscopy (Figure 3d).

## 9. Discussion

ET can occur in immunocompromised or immunocompetent hosts. If the patient has a history of TB exposure and symptoms of dysphagia or upper gastrointestinal bleeding, ET should be considered, especially when: (1) mid-esophageal ulcerations, protuberant lesions, or fistulas are detected with EGD; (2) EUS shows a thickened esophageal wall with destruction. Mediastinal lymph nodes were primarily low echo mixed with high echo, adventitia blurred, mutually fused, or had a close relationship with the esophagus; (3) hypo-enhancement is revealed by CH-EUS; and (4) elastography reveals that the lesions are soft.

Although the morphological specificity of ET is poor according to EGD, other techniques could make up for this defect. EUS is expected to improve the differential diagnostic accuracy of ET from carcinoma and SMTs. In addition, CH-EUS and elastography are able to provide supplementary information. Certainly, diagnosis of ET ultimately depends on pathological and pathogenic evidence. Esophageal mucosal tissue can be acquired by biopsy forceps. Deep biopsy tissue such as infected lymph nodes can be obtained with the aid of EUS-FNA. Endoscopic techniques can also be used to detect multidrug-resistant mycobacterium tuberculosis and close the fistula with OTSC and APC.

Due to the low incidence of ET and the fact that CH-EUS has not been widely used in clinical practice, the deficiency of this article lies in its sample size being inadequate. As the sample size expands, there is a posssibility that ET lesions could manifest as iso- or hyper-enhanced according to CH-EUS. Mechanically, the blood supply of Stage I lymph nodes was normal or even abundant compared with the surrounding tissues. Nonetheless, we speculate that hypo-enhancement accounts for the majority because most patients have progressed to Stage II or a more severe stage when they first seek medical treatment. We need to expand the sample size and study the role of CH-EUS in differential diagnosis of ET from other diseases in the near future.

Endoscopists could protect themselves from the following aspects when encountering suspected ET patients. First, the endoscopy center should be divided into clean area or contaminated area. It is necessary to reduce the movement of people and instruments between different areas. Second, endoscopists should wear caps, protective clothes, masks, double gloves and shoe covers. Finally, disposable accessories should be used whenever possible. Items that must be reused should be strictly sterilized. The environment should also be sterilized after the endoscopic procedure.

Future investigations of ET could focus on the following topics. First, future studies should focus on describing more in detail EUS and CH-EUS features of ET and their potential differentiation from carcinoma, SMTs, sarcoidosis, Crohn’s disease, and Behçet’s disease. Second, the pathological and pathogenic detection method of higher sensitivity and specificity needs to be developed. Third, the therapeutic effects of endoscopy in ET, such as the closure of a fistula, deserve further exploration. In conclusion, endoscopic techniques play crucial roles in the diagnosis of ET. The development of endoscopic techniques such as CH-EUS is expected to improve the diagnostic accuracy of ET.

## Figures and Tables

**Figure 1 jcm-11-07009-f001:**
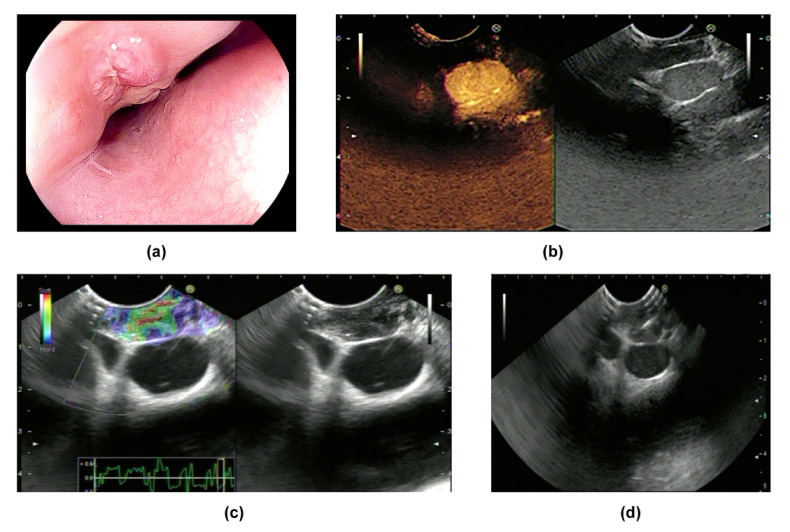
Manifestations of Case 1: (**a**) manifestations of the lesion according to EGD; (**b**) the lesion was hypo-enhanced according to CH-EUS; (**c**) the lesion was soft according to elastography; (**d**) EUS-FNA for the lesion.

**Figure 2 jcm-11-07009-f002:**
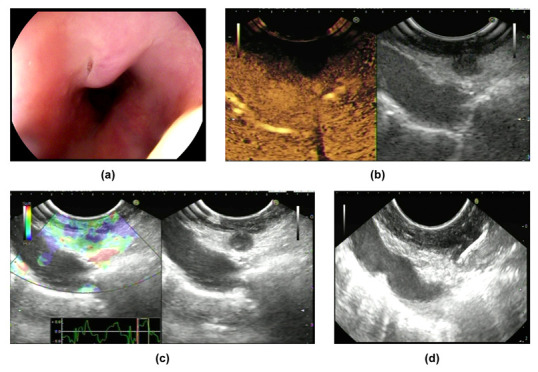
Manifestations of Case 2: (**a**) manifestations of the lesion according to EGD; (**b**) the lesion was hypo-enhanced according to CH-EUS; (**c**) the lesion was soft according to elastography; (**d**) EUS-FNA for the lesion.

**Figure 3 jcm-11-07009-f003:**
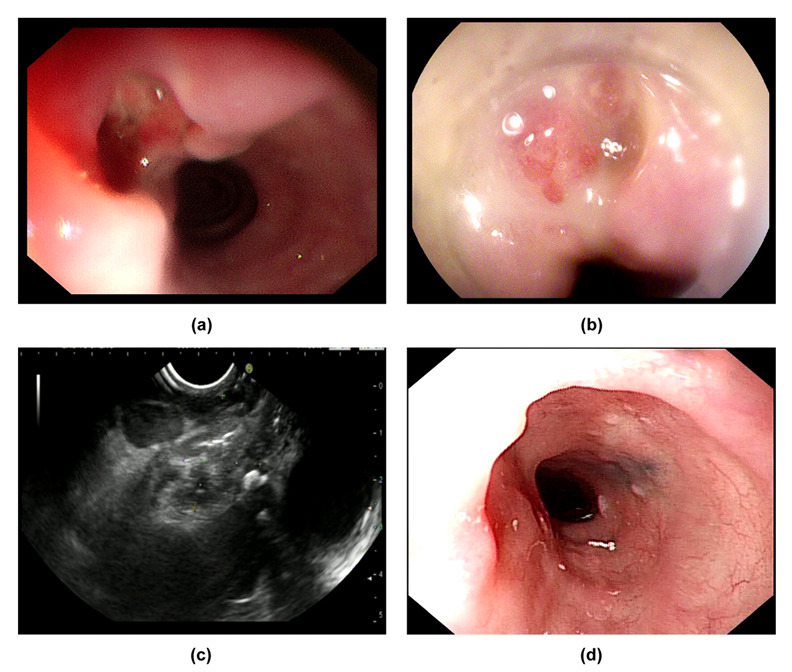
Manifestations of Case 4: (**a**,**b**) manifestations of the lesion according to EGD; (**c**) the lesion as shown by EUS; (**d**) changes shown by using endoscopy after drug treatment.

**Table 1 jcm-11-07009-t001:** Differences between primary and secondary esophageal tuberculosis.

Classification	Primary Esophageal Tuberculosis	Secondary Esophageal Tuberculosis	Reference
Frequency	Rare	Common	[18]
Extraesophageal tubercular lesions	Absence	Present	[13]
Etiology	Swallow sputum or food contaminated with tubercle bacilli	Invasion of adjacent extraesophageal tubercular lesions	[3,14,15,16,17,18]

**Table 2 jcm-11-07009-t002:** Role of different types of endoscopies in the diagnosis of esophageal tuberculosis.

Types of Endoscopy			Features of Esophageal Tuberculosis	Reference
EGD			Ulcerations, eminence lesions, fistula, stricture, traction diverticula	[5,6,17,18,21,22,28,33,34]
EUS	Esophageal wall		Thickening esophageal wall with vague boundary	[17,18]
	Tuberculosis Lymph node	Phase I	Homogeneous and hypoechoic mass	[1,6,17,18]
		Phase II	Heterogeneous which presents as hyperechoic foci or strands on hypoechoic background	
		Phase III	Heterogeneous hyperechoic mixed hypoechoic mass with indistinct adventitia, occasionally partially matting	
CH-EUS			Hypo-enhancement compared with the surrounding tissues	
Elastography			Green	

EGD: esophagogastroduodenoscopy; EUS: endoscopy ultrasonography; CH-EUS: contrast-enhanced harmonic endoscopic ultrasonography.

**Table 4 jcm-11-07009-t004:** Endoscopic treatment of esophageal tuberculosis.

Endoscopic Treatment	Function	Reference
Biopsy forcep/EUS-FNA	Acquire biopsy specimens for multidrug-resistance detection	[57,58,59]
Over-the-scope clip	Closure of fistula	[60,61,62,63]
Argon plasma coagulation	De-epithelialization prior to closure	[64]

EUS-FNA: endoscopic ultrasound-guided fine-needle aspiration.

**Table 5 jcm-11-07009-t005:** Case summary.

No.	Age	Sex	EGD	EUS	CH-EUS	Elastography	Histopathology	PCR/Ziehl-Neelsen Staining
1	28	Female	Eminence	Hypoechoic with hyperechoic	Hypo-enhancement	Soft	Granuloma	Negative/Negative
2	67	Male	Eminence	Hypoechoic	Hypo-enhancement	Soft	Granuloma	Negative/Negative
3	63	Female	Ulcerated stricture	-	-	-	Granuloma	-/-
4	68	Male	Fistula	Hypoechoic with hyperechoic	-	-	Granuloma	-/-

EGD: esophagogastroduodenoscopy; EUS: endoscopic ultrasonography; CH-EUS: contrast-enhanced harmonic endoscopic ultrasonography; PCR: polymerase chain reaction.

## Data Availability

Not applicable.

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
