# Peer review of "Role of Endoscopy in Esophageal Tuberculosis: A Narrative Review"

_jcm, 2022, doi:10.3390/jcm11237009_

Round 1

Reviewer 1 Report

The authors tried to summarize endoscopic findings of esophageal tuberculosis.

1. English editing is needed. 

2. As mentioned in the article, esophageal tuberculosis is very rare. It is classified as primary and secondary. It would be clearer to readers if table summarizing differences is shown.

3. Could you provide more endoscopic pictures showing different features of esophageal tuberculosis? You have given description of ulcers and protrusions, but the descriptions are too general and simple. It would help readers if pictures are provided.

4. The title is "role of endoscopy in the diagnosis and treatment of esophageal tuberculosis: a review", but the article focuses more on diagnosis. Although treatment method is mentioned in the article it would be better to include what changes (such as scar) can be found during endoscopy after treatment of esophageal tuberculosis.

5. Figure legend is not complete. figure legend fo 1c, and 1d are missing.

Author Response

Dear Reviewer,

Yours sincere.

Reviewer 2 Report

Overall, I consider this to be a well-written review for the rare disease ET. Please consider the following points.

1. Since this is a narrative review, please write it as such in the title.

2. The subject of this special issue is "EUS diagnosis and treatment". This submitted paper clearly describes EUS, but I think it would be better if the items were more focused on EUS. In addition, the title should be changed to "role of endosonography" instead of "role of endoscopy".

3. Since EUS is not used for treatment, I think that it is sufficient to describe it briefly.

Author Response

Dear  Reviewer,

Yours sincere.

Reviewer 3 Report

Overall, the whole structure of this study is good. However, some corrections are recommended for providing clear information. Particularly, I listed the following comments in detail here.

Major concerns:

In the introduction, the sentences are un-regularly disperse. Additionally, all of the names and terms should be completely mentioned for the first time in text, such as GI tract, EUS, FNA, and so on.

The references are missed for some sentences. For example, “However, the diagnosis of ET remains challenging because its clinical features are often nonspecific and variable. It may be mistaken for esophageal carcinoma and may lead to unnecessary surgery.”, “Despite the optimistic prognosis of ET, it could become life-threatening in some cases such as massive bleeding resulting from esophagoaortic fistula. Therefore, 37 prompt and explicit diagnosis has always been the essential and arduous topic for ET.”, “Gastroscopy could observe the morphology of the lesion directly. EUS could detected the echogenicity characteristic of the lesion, different layers of esophageal wall and para-esophageal organs and tissues.” and so on.

I propose to add a Table to overview role and Medication and Endoscopic treatment in Esophageal tuberculosis (ET).

In the end, what is your conclusion? Hence, add a significant statement that must be structured as “what was offered by authors? Do the authors have more thoughts on this field?

Author Response

Dear  Reviewer,

Yours sincere.

Round 2

Reviewer 1 Report

The article looks much better after major improvements.

Just one minor comment. The endoscopic images of figure 3a and 3b are too blurry.
